# CONTEXTUAL DOCUMENT EMBEDDINGS

**John X. Morris**
Cornell University
jxm3@cornell.edu

**Alexander M. Rush**
Cornell University
arush@cornell.edu

## ABSTRACT

Dense document embeddings are central to neural retrieval. The dominant paradigm is to train and construct embeddings by running encoders directly on individual documents. In this work, we argue that these embeddings, while effective, are implicitly out-of-context for targeted use cases of retrieval, and that a document embedding should take into account both the document and neighboring documents in context – analogous to contextualized word embeddings. We propose two complementary methods for contextualized document embeddings: first, an alternative contrastive learning objective that explicitly incorporates document neighbors into the intra-batch contextual loss; second, a new contextual architecture that explicitly encodes neighbor document information into the encoded representation. Results show that both methods achieve better performance than biencoders in several settings, with differences especially pronounced out-of-domain. We achieve state-of-the-art results on the MTEB benchmark with no hard negative mining, score distillation, dataset-specific instructions, intra-GPU example-sharing, or extremely large batch sizes. Our method can be applied to improve performance on any contrastive learning dataset and any biencoder.

## 1 INTRODUCTION

Machine learning approaches to text retrieval aim to learn an embedded representation for indexing documents. Classically, this area was dominated by statistical approaches using sparse lexical matching methods based on n-gram frequencies such as BM25 (Robertson & Zaragoza, 2009). Only recently have neural networks become competitive with state-of-the-art models on retrieval tasks (Karpukhin et al., 2020; Thakur et al., 2021). The primary neural method is a *dual encoder* architecture that independently encodes both a document and query to a dense latent space for retrieval lookup. This document embedding space can improve upon a statistical model since it is learned end-to-end for retrieval.

However, there is at least one notable benefit of statistical approaches that is lost by neural models. Statistical models can easily incorporate prior corpus statistics such as inverse document frequency (IDF), into their representation. This prior term imparts context-dependence onto the model, since it can be updated based on information specific to retrieval in a given domain at test time. We contrast this contextual formulation with neural document encoders that are by definition a function of the document itself. For example consider the following document:

> The National Football League Draft is an annual event in which the National Football League (NFL) teams select eligible college football players...

Depending on the retrieval domain, e.g. Wikipedia search, sports articles, or televised events, IDF may weight terms such as `NFL`, `draft` or `annual` higher; a neural document embedding model would need to select a global weighting for this document.

In this work, we explore contextualization of document embeddings produced by dense encoders. The goal is to produce embeddings that are better able to handle retrieval tasks in specific challenging contexts. We propose two complementary changes to document encoders: a contextual training procedure and architecture.

For contextual training, we aim to build a notion of neighboring documents directly into the contrastive learning process. We propose a method that uses fast query-document clustering to produce a group

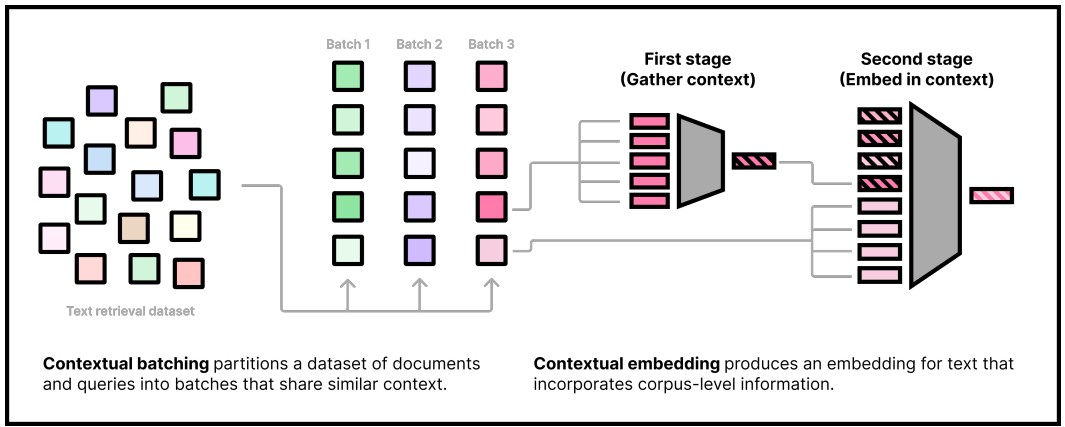

Figure 1: Overview of our system for contextual document embeddings (CDE). Our model operates in two stages: a first stage used to characterize the dataset from samples, and a second stage used to embed the final document.

of neighbors for each training batch. Each update for training is constructed purely from neighboring documents to ensure that embeddings can distinguish documents even in the most challenging contexts.

For the architecture, we propose a new encoder that injects information about the contextual documents during embedding. The proposed architecture augments the standard BERT-style encoder with additional conditioning that provides aggregated document-level information about neighboring documents. We call our method Contextual Document Embedding (CDE). Analogously to pre-computed corpus-level statistics, this method provides a manner for the embedding to take into account the relative frequency of terms in context. The final output is still an embedding of the same size, so this does not require any additional storage or other changes to the retrieval process. When indexing, we utilize information from the corpus to produce document and query embeddings that are specific to a particular domain.

Experiments compare these two extensions to standard approaches for training document embeddings. Our results show that contextual contrastive training improves standard text embedding model training, and can be run without other approaches such as additional hard negatives. With the contextual encoder architecture, we see additional improvements over a baseline model in all settings tested, with larger improvements in highly specific domains such as small datasets of financial and medical documents. When trained at industry-scale, our model achieves state-of-the-art results for small ($<$250M parameter) models on the MTEB benchmark.

## 2 RELATED WORK

**Text retrieval.** Our work is related to the general field of text retrieval; we propose specific improvements to the training of "biencoder" text embedding models such as DPR (Karpukhin et al., 2020), GTR (Ni et al., 2021), Contriever (Izacard et al., 2022), LaPraDoR (Xu et al., 2022), Instructor (Su et al., 2023), Nomic-Embed (Nussbaum et al., 2024), E5 (Wang et al., 2024), and GTE (Li et al., 2023). We focus on the problem of adapting these text retrieval models to new corpora at test time; some prior work has noted this problem (Dai et al., 2022; Sciavolino, 2021) and proposed solutions such as unsupervised span-sampling and training on test corpora (Gao & Callan, 2021) and distillation on the test corpus from a reranker (Sung et al., 2023). Late interaction methods (Khattab & Zaharia, 2020; Santhanam et al., 2022) also offer one way to improve out-of-domain retrieval performance, but increase the runtime and complexity of search. We propose a better sampling scheme that can be used to train any biencoder or late interaction model as well as a *training-free* method for test-time adaptation.

**Contrastive learning.** Much research has focused on the effect of hard negatives on the performance of contrastive learning methods Chen et al. (2020); Qu et al. (2021); Robinson et al. (2021); Wang et al. (2023). (Zhang & Stratos, 2021) observe that harder negatives provide a better approximation of the overall cross-entropy loss, but do not consider *batch*-level optimizations for negative selection. Hofstätter et al. (2021) cluster queries before training and show that this improves performance. Sachidananda et al. (2023) also consider contrastive batch sampling as a global optimization problem, but do not apply their technique to state-of-the-art transformer-based text embedding models. (Ma et al., 2024) use a clustering algorithm to partition a dataset into several sub-datasets, but train a different model on each sub-dataset. Solatorio (2024) also use a pre-trained model to address the problem of in-batch false negatives from randomly sampled batches. Our training algorithm aims to find the hardest possible high-quality batches to train text embedding models.

**Test-time adaptation.** Our method can be compared to other solutions to test-time adaptation, a problem that has been well-studied across a variety of domains (Jang et al., 2023). In retrieval, one form of test-time adaptation is pseudo-relevance feedback (PRF) (Rocchio, 1971; Li et al., 2018; Wang et al., 2021), where documents relevant to the query are used to construct a final, enhanced query representation. The query side of our model can be seen as a form of pseudo-relevance feedback; however, we train from scratch to support a more general form of PRF natively, on the document representation as well as the query.

**Non-parametric modeling.** Our contextual document model can be seen as a form of non-parametric modeling. This shows connections with the a large body of deep learning research such as the non-parametric transformer (NPT) (Kossen et al., 2022) and the subfield of Neural Processes (Garnelo et al., 2018; Kim et al., 2019; Nguyen & Grover, 2023). Semi-parametric models have been recently applied in NLP, specifically to the task of language modeling (Borgeaud et al., 2022; Khandelwal et al., 2020). Instead of using a retrieval model to build a semi-parametric langauge model, we build a semi-parametric model specifically for the task of retrieval.

## 3 BACKGROUND

We can view text retrieval methods probabilistically as computing a distribution over potential documents based on a scalar score function $f(d, q)$ matching documents and queries:

$$p(d \mid q) = \frac{\exp f(d, q)}{\sum_{d' \in \mathcal{D}} \exp f(d', q)} \tag{1}$$

where $\mathcal{D}$ is a finite set of documents in a dataset. There is a wide variety of different definitions for $f$ including full pairwise neural parameterizations (Nogueira & Cho, 2020). In this work, we focus on efficient retrieval methods using vector-based methods, also known as embedding models.

Vector retrieval methods assume that $f(d, q)$ can be factored into two embedding terms, $\phi(d) \cdot \psi(q)$, the document and query embedding respectively. This factorization allows precomputation of the document embeddings $\phi(d)$ for all $d \in \mathcal{D}$. This is critical for facilitating fast computation of $\arg\max_d p(d \mid q)$ or top-k variants (Douze et al., 2024).

In statistical retrieval, $\phi$ and $\psi$ are closed-form functions of the data, often representing unigram or bigram counts by the relative frequency of word types. Notably for this work, these methods can also utilize distributional properties of the test dataset as a prior, for example through inverse document frequency (IDF). We represent this integration of dataset-level information by writing the vector product $\phi(d; \mathcal{D}) \cdot \psi(q; \mathcal{D})$.

In neural retrieval, we instead learn the representation as a dense vector. We assume access to a training corpus of document and query pairs (these may be supervised, i.e. gold-standard annotations, or unsupervised, i.e. noised synthetic examples), $\mathcal{D}_T = \{(d^1, q^1), ..., (d^J, q^J)\}$, with the aim of learning the embedding function $\phi$ and $\psi$.

Training can be motivated as maximizing likelihood of the document corresponding to each query, i.e. $\sum_j \log p(d^j \mid q^j)$. Unfortunately, since retrieval datasets can have $|\mathcal{D}|$ exceed millions of documents, computing the normalizer in Eq 1 at each training step is not an option. Instead contrastive learning is used where the likelihood is replaced with a biased approximation calculated from negative samples:

$$\max_{\phi,\psi} \sum_j \log p(d^j \mid q^j) \approx \sum_j \log \frac{\exp f(d^j, q^j)}{\sum_{d' \in \mathcal{H}(q^j)} \exp f(d', q^j)}$$

where $\mathcal{H}$ is a set of examples used to approximate the normalizing constant. In implementation, in addition to these hard negative examples, other examples from the mini-batch are also used to compute the normalizer since it requires no additional compute for calculating $\phi(d)$.

## 4 METHODS

In our work, we are interested in integrating contextual information into our embedding functions $\phi$ and $\psi$. The standard neural $\phi$ is purely a function of the document $\phi(d)$ and does not take into account any notion of context. This contrasts with the statistical model $\phi(\cdot; \mathcal{D})$ and $\psi(\cdot; \mathcal{D})$. Arguably this is not an issue if retrieval is completely in domain, as $\phi$ is capable of learning statistics such as IDF and average document length on the training set through gradient descent.

However, in many retrieval benchmarks, models are trained over a single set of documents $\mathcal{D}$ and then tested in many other domains $\mathcal{D}$ that differs significantly from $\mathcal{D}_T$. In this setting, training on $\mathcal{D}_T$ alone may not be able to provide robust embeddings when used in contexts such as $\mathcal{D}$.

### 4.1 CONTEXTUAL TRAINING WITH ADVERSARIAL CONTRASTIVE LEARNING

Returning to the example from the introduction, we assume that in a general purpose training corpus $\mathcal{D}_T$, the term NFL is a rare word appearing in relatively few documents and a useful signal. However, if at test time $\mathcal{D}$ is a corpus of sports articles, this word would be exceedingly common. Evaluation in this domain is, in a statistical sense, adversarial to the original dataset. To handle this issue, meta-learning-style objectives have shown to be effective for training document embedders. In these approaches, instead of sampling documents-query pairs iid, the objective first sample a domain and then sample a batch of examples. This ensures that the model mostly sees related training points in each domain.

We propose a training objective that synthesizes a large set of fine-grained domains to train the model on. Formally, our aim is to partition the training dataset $\mathcal{D}_T$ into groups $(\mathcal{B}^1, \dots \mathcal{B}^B)$ such that each group represents a self-similar pseudo-domain:

$$\max_{\phi,\psi} \sum_b \sum_{(d,q) \in \mathcal{B}^b} \log p(d \mid q) = \max_{\phi,\psi} \sum_b \sum_{(d,q) \in \mathcal{B}^b} \log \frac{\exp f(d, q)}{\sum_{(d', \cdot) \in \mathcal{B}^b} \exp f(d', q)}$$

Computationally, the inner term can be implemented as a single batch and computed efficiently without the need for separate hard negatives ($\mathcal{H}$). Ideally we want groups that are as challenging as possible. Zhang & Stratos (2021) show that increasing the partition term improves the contrastive approximation to the maximum likelihood of the gradient. We can formalize this search for the most difficult configuration of batches as an optimization problem:

$$\max_{(\mathcal{B}^1, \dots \mathcal{B}^B)} \sum_b \sum_{\substack{(d,q) \in \mathcal{B}^b \\ (d',q') \in \mathcal{B}^b}} (f(d, q') + f(d', q)) = \max_{(\mathcal{B}^1, \dots \mathcal{B}^B)} \sum_b \sum_{\substack{(d,q) \in \mathcal{B}^b \\ (d',q') \in \mathcal{B}^b}} (\phi(d) \cdot \psi(q') + \phi(d') \cdot \psi(q))$$

$$(2)$$

Solving this combinatorial objective exactly is intractable, but we can approximate a solution using clustering. We first move from a maximization to a minimization by replacing the two dot products with $L_2$ distance $M((d,q), (d', q')) = ||\phi(d) - \psi(q')|| + ||\phi(d') - \psi(q)||$ (which is equivalent for normalized embeddings). We then note when that treated as symmetric pairs, this term obeys the triangle inequality for any other centroid pair $c$:

$$M((d, q), c) + M(c, (d', q')) \geq M((d, q), (d', q'))$$

This implies that the following centroid-based objective represents an upper-bound on our original objective:

$$\min_{\substack{(\mathcal{B}^1,...\mathcal{B}^B) \\ (m^1,...,m^B)}} \sum_b \sum_{(d,q)\in\mathcal{B}^b} M((d,q),c^b) \tag{3}$$

For known $B$, this search defines an asymmetric K-Means clustering problem. A solution can be efficiently computed using extremely fast Euclidean K-Means packages be treating each data point as two separate vectors $\phi(d) \oplus \psi(q)$ and $\psi(q) \oplus \phi(d)$, where $\oplus$ is concatenation.

**Cluster Embeddings.**    Since clustering is performed before training, we do not have dense encoders $\phi$ and $\psi$ when constructing the groups. Borrowing methods from hard-negative mining (Robinson et al., 2021) we can replace the $\phi$ and $\psi$ with a simpler embedding model when constructing groups. We experiment with a sparse vector representation and with pretrained dense representations, settling on GTR (Ni et al., 2021), a popular and generic text embedding model.

**Filtering False Negatives.**    Our method is especially sensitive to false negatives, as they will be more likely to be included in a given batch. Unfortunately, traditional retrieval datasets are not designed with this type of global objective in mind: false negatives are common in most retrieval datasets and their prevalence increases with dataset scale. As one datapoint, Qu et al. (2021) found that over 70% of top-retrieved passages in MS Marco are false negatives.

To avoid a situation where each batch contains a large number of false negatives, we compute an equivalence class: $S(q,d) = \{d' \in \mathcal{D} \mid f(q,d') \geq f(q,d) + \epsilon\}$ for some surrogate scoring function $f$ and boundary term $\epsilon$. At training time, we alter the partition function for $d$ so that it no longer includes the elements of $S(q,d)$, which are not definitively negative examples:

$$\log p(d \mid q) = \frac{\exp f(d,q)}{\exp f(d,q) + \sum_{d' \notin S(q,d)} \exp f(d',q)} \tag{4}$$

For simplicity, we again select $f$ to be a simple pre-trained embedding model. This method likely over-prunes some potential true negatives found by the surrogate model; however we found it to be critical to model accuracy.

**Packing.**    Clusters found by our algorithm will be of varying sizes, and need to be packed into equal-sized batches. We apply a post-hoc procedure. We consider both random partitioning and grouping via greedy cluster-level traveling salesman, similar to Shi et al. (2024). In both cases, we split large group into into smaller batches, and merge close small batches from within the same domain into evenly-sized batches. This has an added benefit of introducing randomness into the groups when training for multiple epochs. We leave it to future work to analyze the full effects of different packing strategies such as expensive Balanced K-Means or heuristic approaches such as Equal K-Means (Gururangan et al., 2023).

## 4.2 CONTEXTUAL DOCUMENT EMBEDDING (CDE)

Contextualization can also be added directly to the architecture. Taking inspiration from sparse vector retrieval which uses corpus statistics to determine the form of the embedding, we modify the encoders to have access to the corpus itself, i.e. $\phi(d; \mathcal{D})$ and $\psi(d; \mathcal{D})$. This effectively augments the biencoder model to give it the ability to contextualize documents directly.

The main challenge is how to design a neural architecture that can take into account dataset contextualization. On one extreme, we could follow methods like BM25 and precompute a fixed set of corpus statistics that could be fed to the document encoder. On the other extreme, we could allow the encoder full access to the entire corpus, through some form of cross attention. The latter approach has been explored on a small scale in methods like neural processes (Garnelo et al., 2018); however, it has been shown to be difficult to scale to larger datasets Borgeaud et al. (2022).

We opt for a middleground that allows the model to learn corpus statistics, but is also relatively efficient to compute, shown in Figure 1. Specifically, we note that document embeddings retain a surprising amount of lexical information even after embedding (Morris et al., 2023). Therefore, if we pre-embed a subset of the corpus, we believe we can still dynamically calculate key dataset information during encoding.

We produce contextualized embeddings via a two-stage process:

**First stage:** *Gather and embed context.* Given context documents $d^1, ..., d^J \in \mathcal{D}$, we embed each using a unique embedding model and concatenate embeddings into a sequence $M_1(d^1)...M_1(d^J)$.

**Second stage:** *Embed document with additional context tokens.* To compute $\phi$ for document $d'$ we integrate contextual embedding sequence at the input of second-stage embedding model $M_2$:

$$\phi(d'; \mathcal{D}) = M_2(M_1(d^1), \ldots, M_1(d^J), E(d'_1), \ldots, E(d'_T)) \tag{5}$$

Here $M_1$ is the first-stage encoder model, $M_2$ is a second-stage encoder model, and $E$ is the token embedding matrix of $M_2$ applied to each token in $d'$. In practice, we parameterize both $M_1$ and $M_2$ using traditional bidirectional transformers, so our model is comprised of two biencoder-like backbones called in sequence.

There is a similar contextualized model for the query encoder $\psi$ which is also given document context (as we do not have query context at test time):

$$\phi(q; \mathcal{D}) = M_2(M_1(d^1), \ldots, M_1(d^J), E(q_1), \ldots, E(q_T)) \tag{6}$$

We note several implementation properties of this architecture. During training, computing contextual embeddings for each contextual document for each training instance would naively increase training by a computational factor proportional to $J$, the number of documents in context. This time increase would not be tractable, since contrastive training can already take many days. We overcome this difficulty by sharing context $d^1, ..., d^J$ within a batch of documents; this allows us to compute representations just once per training step and reuse them between documents via computational graph. [1]

When indexing a new corpus $\mathcal{D}$, first stage representations $M_1(d^1)...M_1(d^J)$ can be computed once and cached, so $M_1$ does not add parameters or runtime to the search process; only $M_2$ needs to be stored and used at search time. Query representations can also use the cached context, which only require additional inputs to the encoder. (Our model does not include contextualized queries, only documents, as we typically do not assume access to example queries at test-time.)

**Embedding *without* context.** Individual corpora during training may not have sufficient or available context. To improve our model's generalization, we use *sequence dropout*, where we randomly replace context embeddings $M_1(d^*)$ with some null token $v_\emptyset$ according to some a uniform probability $p$.

At test time, if no corpus information is available, our model can now function as a non-contextual biencoder simply by replacing all sequence token inputs with $v_\emptyset$.

**Position-agnostic embedding.** Since documents of $\mathcal{D}$ are unordered, we remove all positionality from the neural encodings. When parameterizing $\theta$ with a traditional transformer, this can be achieved by omitting positional embeddings at the positions corresponding to $\mathcal{D}$. In practice, we use transformers implementations dependent on FlashAttention with rotary positional embeddings at each self-attention layer. Full details of how we disable positionality are available in Section 10.4.

**Two-stage gradient caching.** To improve training we employ a gradient-caching technique analogous to a two-stage version of GradCache (Gao et al., 2021). This technique allows us to fit larger batches, longer sequences with more contextual samples without running out of memory. Essentially, we compute first-stage and second-stage representations independently without gradients. We then use these frozen representations to compute the loss, and gradients with respect to the second-stage representations. We then re-run the second stage with gradients enabled and use the output gradients to backpropagate through the second-stage model, and obtain gradients for the first-stage representations. We repeat this process for the first-stage representations. This allows us to tradeoff computation (running each transformer forward pass twice) for memory.

---

[1]Context reuse is only feasible because documents within the same batch typically share a large amount of context anyway, since they are clustered.

## 5 EXPERIMENTAL SETUP

We consider a range of retrieval experiments across different scales. To run experiments across a suitable number of settings, we devise a small setting: six-layer transformer, maximum sequence length of 64, and maximum number of 64 additional contextual tokens. In this scenario, we evaluate on a truncated version of the BEIR benchmark (Thakur et al., 2021). Given the low cost of each experiment, we are able to pre-train and fine-tune both biencoder and contextual models across a variety of batch sizes in $\{256, 512, 1024, 2048, 4096\}$ and cluster sizes $\{64, 256, 1024, 4096, ..., 2097152, 4194304\}$. As typical state-of-the-art text embedding models are trained in two phases, a large weakly-supervised pre-training phase and a short supervised phase, we run all experiments for both phases.

For the large setting, we use the best settings found via small experiments. We train a single model on sequences of length 512 with 512 contextual documents, evaluating on the full MTEB benchmark (Muennighoff et al., 2022). This includes tasks from retrieval as well as tasks like classification, clustering, and reranking. We compare our model's performance to the top small-size (under 250M parameters) models on MTEB (Nussbaum et al., 2024; Xiao et al., 2024; Solatorio, 2024; Li et al., 2023).

**Training Data and Metrics** We train on the meta-datasets collected in Nussbaum et al. (2024) for training text embedding models. This collection of datasets includes data from 24 datasets scraped from web sources such as Wikipedia and Reddit. Our unsupervised training phase trains on 200M weakly-supervised datapoints scraped from large internet sources such as Reddit and Wikipedia. The supervised training phase includes 1.8M human-written query-document pairs intended for text retrieval, and is aggregated from popular retrieval datasets such as HotpotQA and MS MARCO (Yang et al., 2018; Bajaj et al., 2018). For our full model, we also consider supervised training on the BGE meta-datasets (Xiao et al., 2024). We evaluate our models using NDCG@10, a conventional retrieval metric that enables comparison across many disparate datasets.

**Implementation** When partitioning our dataset into batches, we encode documents and queries using GTR (Ni et al., 2021) and implement our clustering algorithm on top of FAISS (Douze et al., 2024). We cluster per-domain for 100 steps and take the best clustering out of 3 attempts. We select NomicBERT as our pre-trained model backbone (Nussbaum et al., 2024), which has 137M parameters. We prepend all texts with short task-specific prefixes to identify each task; prefixes are listed in Section 10.7. When pooling, we pool over text tokens only, never contextual tokens.

**Training** We initialize both $M_1$ and $M_2$ using the BERT-base model from Nussbaum et al. (2024) that includes flash attention. Weights are shared between $\phi$ and $\psi$, but notably not between $M_1$ and $M_2$. For all experiments, we train with the Adam optimizer with 1000 steps of warmup to a learning rate of $2 \cdot 10^{-5}$ and linearly decay to 0 throughout training. For the filtering model we select `nomic-embed-v1` which was trained on the same datasets (Nussbaum et al., 2024). We train for three epochs unless otherwise specified. We set the maximum sequence length for all inputs to 512 and the number of contextual inputs to 512 (so the second-stage model has an input length of 1024). When computing contrastive loss, we use a fixed temperature of $\tau = 0.02$. When sequence dropout is enabled in our contextual architecture, we set contextual input tokens to null vectors with a uniform probability $p = 0.005$. If the batch size exceeds the number of contextual documents, we randomly sample to produce contextual inputs.

## 6 RESULTS

The main results are highlighted in Table 1 and Section 6. In the smaller setting, we observe that both adversarial contrastive learning and our contextual architecture improve performance compared to vanilla biencoder training. We observe the largest improvement when we combine these techniques.

**Contextual batching** After controlling for batch size and filtering for false negatives, we observe a strong correlation (visualized in Figure 13) between batch difficulty and downstream performance: *reordering datapoints to make batches harder definitively enhances overall learning*. This corroborates prior findings (Xiong et al., 2020; Qu et al., 2021) and theory (Zhang & Stratos, 2021) that

| Contextual | | | | | | |
|---|---|---|---|---|---|---|
| Batch | Arch | Batch Size | Cluster Size | Train loss | Train acc. | NDCG@10 |
| | | 16384 | - | 0.39 | 90.3 | 59.9 |
| ✓ | | 512 | 512 | 0.81 | 77.7 | 61.7 |
| | ✓ | 16384 | - | 0.37 | 90.7 | 62.4 |
| ✓ | ✓ | 512 | 512 | 0.68 | 80.9 | **63.1** |

Table 1: Performance of our small models with and without the two improvements proposed in this paper, measured on a shortened version of the BEIR benchmark. Numbers are NDCG@10.

more difficult batches in contrastive learning form a better overall gradient approximation and learn more effectively.

Section 10.12 showcases model performance across batch and cluster sizes after both phases of training. We observe that although a large batch and cluster size are useful when filtering is not enacted, when including filtering, smaller cluster (and harder) are clearly better, and large batches do not add much. When comparing filtered to non-filtered models (Figure 15), filtering false negatives clearly improves performance.

**Contextual architecture** In addition to adversarial batching, we compare our contextual architecture to a biencoder across the datasets of BEIR in Table 1 (full results in appendix). Our architecture generally matches or improves performance on all downstream datasets, with largest improvements in ArguAna and ClimateFEVER, two of the smaller and more out-of-domain datasets.

**Full-scale training** Figure 16 shows our models' performance when trained for multiple epochs on the supervised datasets, relative to the best similar-sized embedding model (dashed line). We find best performance when training for four epochs on the BGE meta-datasets. Although our best model does use a single hard negative per query, we are still able to to achieve state-of-the-art performance without using *any* hard negatives.

For our final model (`cde-small-v1`), we select the best of the supervised models[2], which comes from finetuning on the BGE dataset. On MTEB, `cde-small-v1` obtains state-of-the-art results compared to models of the same size. Although inspired by problems in the specific domain of text retrieval, we observe that our approach improves embedding performance in all domains, including clustering, classification, and semantic similarity. We also evaluate a "random documents" baseline, where we sample random documents from the training dataset to simulate a scenario where we lack access to the test corpus. In this setting, we drop around 1.2 points on average across all tasks; the STS tasks in particular appear to produce representations that are close to context-agnostic.

| Method | Arg | CQA | CFEVER | DBP | FEVER | FiQA | HPQA | MSMRC | NFC | NQ | QUORA | SCID | SCIF | TREC | TOUCHE | Mean |
|---|---|---|---|---|---|---|---|---|---|---|---|---|---|---|---|---|
| **Unsupervised** | | | | | | | | | | | | | | | | |
| Baseline | 54.8 | 41.4 | 24.7 | 40.2 | 74.4 | 39.9 | 63.8 | 35.0 | 35.7 | 48.6 | 88.2 | 20.2 | 72.0 | 62.2 | 19.2 | 48.0 |
| Contextual | 54.9 | 43.1 | 24.4 | 40.7 | 79.6 | 42.1 | 68.8 | 38.9 | 36.5 | 57.8 | 88.9 | 21.1 | 72.8 | 77.1 | 21.9 | 51.2 |
| **Supervised** | | | | | | | | | | | | | | | | |
| Baseline | 49.3 | 40.5 | 38.3 | 45.0 | 85.0 | 38.4 | 73.6 | 43.1 | 35.0 | 59.4 | 87.7 | 18.3 | 70.5 | 79.9 | 28.2 | 52.8 |
| Contextual | 53.8 | 41.2 | 38.8 | 43.3 | 89.2 | 40.1 | 73.9 | 42.2 | 35.9 | 61.6 | 87.1 | 20.1 | 72.7 | 82.6 | 27.8 | 54.0 |

Table 2: Results (NDCG@10) on the retrieval setting of the MTEB benchmark.

# 7 ANALYSIS

**How hard are our clusters?** To analysis the relationship between cluster size in our clustering algorithm and the overall average difficulty of in-batch negatives, we measure the average difficulty of 1000 batches across a variety of batch and cluster sizes and plot the data in Figure 17. We observe that larger batches bring easier non-negative examples, and decreasing cluster size clearly increases the average hardness of negative examples in a given cluster.

---

[2]We select models by choosing the best batch size and cluster size according to average NDCG on BEIR.

|  | Clssfctn | Cluster | PairCls | Rerank | Retrvl | STS | Summ. | Mean |
|---|---|---|---|---|---|---|---|---|
| nomic-embed-v1 | 74.1 | 43.9 | 85.2 | 55.7 | 52.8 | 82.1 | 30.1 | 62.39 |
| stella-base-en-v2 | 75.3 | 44.9 | 86.5 | 58.8 | 50.1 | 83.0 | 32.5 | 62.61 |
| bge-base-en-v1.5 | 75.5 | 45.8 | 86.6 | 58.9 | 53.3 | 82.4 | 31.1 | 63.56 |
| GIST-Embedding-v0 | 76.0 | 46.2 | 86.3 | 59.4 | 52.3 | 83.5 | 30.9 | 63.71 |
| gte-base-en-v1.5 | 77.2 | 46.8 | 85.3 | 57.7 | 54.1 | 82.0 | 31.2 | 64.11 |
| `cde-small-v1` | | | | | | | | |
| [Random] | 81.3 | 46.6 | 84.1 | 55.3 | 51.1 | 81.4 | 31.6 | 63.81 |
| [Contextual] | 81.7 | 48.3 | 84.7 | 56.7 | 53.3 | 81.6 | 31.2 | **65.00** |

Table 3: Performance of models with 250M or fewer parameters on the MTEB benchmark for text embedding models. "Random" indicates the performance of our model with random training documents included instead of per-domain contextual documents.

|  | Rand. | Cont. | Diff |
|---|---|---|---|
| ClimateFEVER | 19.2 | 25.7 | **+6.5** |
| ArguAna | 66.5 | 72.0 | **+5.5** |
| FiQA2018 | 40.7 | 45.4 | **+4.7** |
| FEVER | 81.6 | 86.1 | **+4.5** |
| BiorxivClusteringP2P | 41.5 | 44.9 | **+3.4** |
| TRECCOVID | 76.1 | 79.4 | **+3.3** |
| MedrxivClusteringP2P | 34.6 | 37.7 | **+3.1** |
| HotpotQA | 63.8 | 66.7 | **+2.9** |
| NQ | 57.1 | 60.0 | **+2.9** |
| SciDocsRR | 80.1 | 82.9 | **+2.8** |

Table 4: Tasks with most improvement when contextual documents are provided. Scores are NDCG@10.

**Which contextual documents help?** To confirm that the CDE model is utilizing contextual information from $\mathcal{D}$ we consider how different contextual documents help for a given docuent $d$. Figure 18 measures results on CQADupstack, a collection of Stack Exchange forum posts. We randomly sample inputs to from $\mathcal{D}$ from a domain (x-axis) and use them as input to the downstream task $d$ marked along the y-axis. We mark a square as red if its score comes within 1 point of NDCG of the top score for its domain. Generally utilizing in-domain works best, but there are some crossover interactions.

## 8 CONCLUSION

We propose two improvements to traditional biencoder models for generating embeddings. The first improvement involves an algorithm for reordering training datapoints to make batches harder and improves vanilla training with minimal changes. Our second improvement involves a new corpus-aware architecture for retrieval and allows us to train a state-of-the-art text embedding model.

## 9 ACKNOWLEDGEMENTS

Thanks to Orion Weller, Vin Sachidananda, and Zach Nussbaum for valuable feedback on this research. We would also like to acknowledge to Nomic and Hyperbolic for providing the compute necessary to conduct this research. This work was partially supported by Intelligence Advanced Research Projects Activity (IARPA), via the HIATUS Program #2022-22072200003. JM is supported by an NSF GFRP fellowship.

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
