# 10    SUPPLEMENTARY MATERIAL

## 10.1    COMPUTATIONAL RESOURCE USAGE

We pre-train all models on 8 NVIDIA H100 GPUs. In the slowest setting, training a biencoder for a single unsupervised epoch (235M pairs) takes approximately one day. Training our contextual archiecture for a single epoch takes approximately two days. Shorter sequence-length experiments are 10-20x faster, and can be run on a single GPU.

## 10.2    INITIAL EXPERIMENTS

We conducted two preliminary experiments to verify (i) the need for contextual training strategy and (ii) the need for in-batch false negative filtering when doing adversarial contrastive learning on a real dataset.

**Preliminary experiment (i).**    We conduct a preliminary experiment to verify this issue. Starting from several trained retrieval systems we compute performance on a variety of different tasks from the BEIR dataset. Additionally we compute the IDF statistics from the datasets, and compare the divergence from the base IDF statistics of the training set. Figure 2 shows that datasets with high-divergence have very high correlation with the accuracy degradation of models when measured in comparison to BM25, which is able to measure and adapt to statistics of the test corpus.

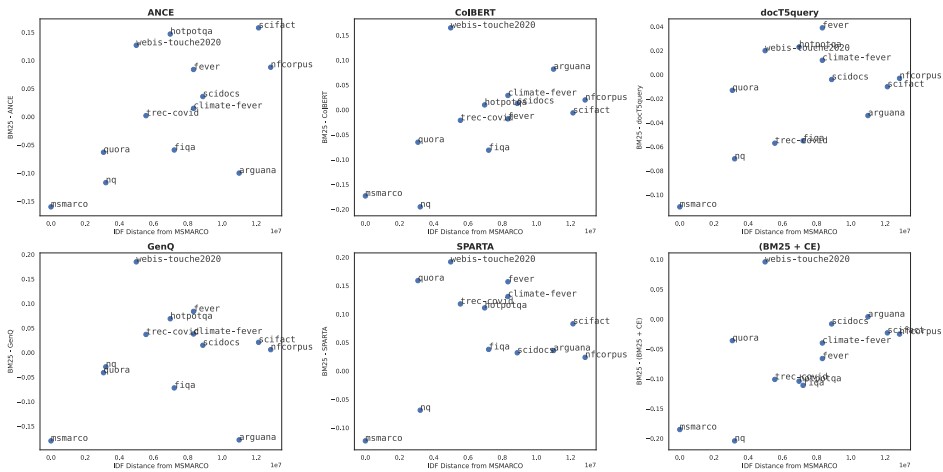

Figure 2:  Analysis of domain shift for popular neural retrieval methods. Performance difference from BM25 (y-axis) correlates with the different in IDF of the test corpus $\mathcal{D}$ form the training corpus $\mathcal{D}_T$.

**Preliminary experiment (ii).**    We select a random document from an unsupervised corpus and look at its nearest neighbors, displayed in Table 5. We observe that the nearest neighbors to a given document in a large corpus are very close; in fact, many of them could be considered valid documents for the given query as well.

## 10.3    INTERACTIONS BETWEEN CONTRASTIVE LOSS AND DISTRIBUTED DATA PARALLEL

The authors note that it can be notoriously difficult to train models using both contrastive loss and the distributed data parallel (DDP) setting. In particular, when aggregating samples between GPUs, if any artifact reveals which GPU a model came from (for example, if the GPU model weights are initialized slightly differently) than the model can quickly deteriorate to a suboptimal solution, each GPU learning a different final model and "cheating" to classify samples based on which GPU they came from.

Table 5: Nearest-neighbors to a single query in a large unsupervised dataset.

| Query | Document |
|---|---|
| looks like my card payment was duplicated after all. [...] | |
| why is there an extra €1 fee in my statement? | why is there an extra charge on my statement? |
| what is this fee for card payment? | why was a fee charged for my card payment? |
| why do i have duplicate transactions for one purchase? | why was my transaction charged twice? |
| i have two of the same charges on my account! | why was my transaction charged twice? |
| my transaction went through but i was charged a fee. why? | why was a fee charged for my transfer? |
| my account shows i have been charged twice for the same meal. [...] | |
| will i get extra charges? | why was a fee charged for my transfer? |
| i got charged in double and want a refund | why was my transaction charged twice? |
| where do i pay with my debit or credit card? | why is my card not accepted? |
| why did i get charged a fee for my card payment? | why was a fee charged for my card payment? |
| my statement shows different transaction times. | why was my transaction charged twice? |

This issue is made extra difficult by the fact that gradient-syncing must be disabled for large-batch contrastive learning to work efficiently. If gradient syncing becomes totally disabled, the training silently diverges as each model learns a degenerate solution. We advise practitioners to take care when controlling gradient-syncing and run many control experiments to determine performance equivalence between DDP and non-DDP scenarios.

One potential benefit of our method is that it greatly decreases the number of hard negatives required per batch, which means that negative-sharing across GPUs may not be necessary in most settings. If possible, the most sanity-preserving way to perform contrastive training could be to

## 10.4 REMOVING POSITIONALITY WITH ROTARY EMBEDDINGS

One detail of our model architecture is that it does not track positionality between dataset input tokens. Although disabling positionality would be trivial an a BERT-like encoder model that uses learned positional embeddings, we use a version of BERT with *rotary* positional embeddings which inject positional information at each layer of the transformer. To circumvent this step, we modify the model internals to set dataset input tokens to zero for the rotary embedding step only, and add a residual connection propagating the dataset input tokens past the rotary embedding phase. Crucially, all contextual tokens are still included for both the MLP and self-attention steps of the transformer.

## 10.5 ADDITIONAL RESULTS

Section 10.5 show sweeps over batch and cluster sizes under our small experimental settings when performing unsupervised pretraining with contextual architecture. We see similar trends to those observed with the biencoder architecture, however we note that performance is higher across the board and our transductive model is able to perform well even at higher cluster sizes and low batch sizes.

One confounding factor in these experiments is that since the number of contextual documents is fixed, the number of different contextual inputs seen during training decreases with higher batch size. This might explain part of why performance stagnates with higher batch sizes; increasing the batch size decreases the total number of learning examples seen by our contextual model.

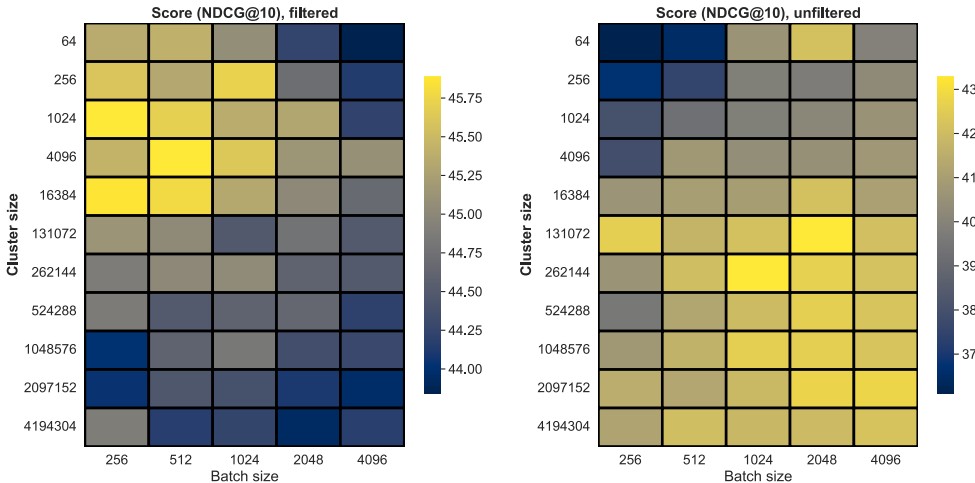

Figure 3: Contextual performance with filtering (left) and without (right) across batch and cluster sizes during unsupervised contrastive pre-training. Here, clustering with small cluster sizes clearly improves performance, and larger batch sizes do not.

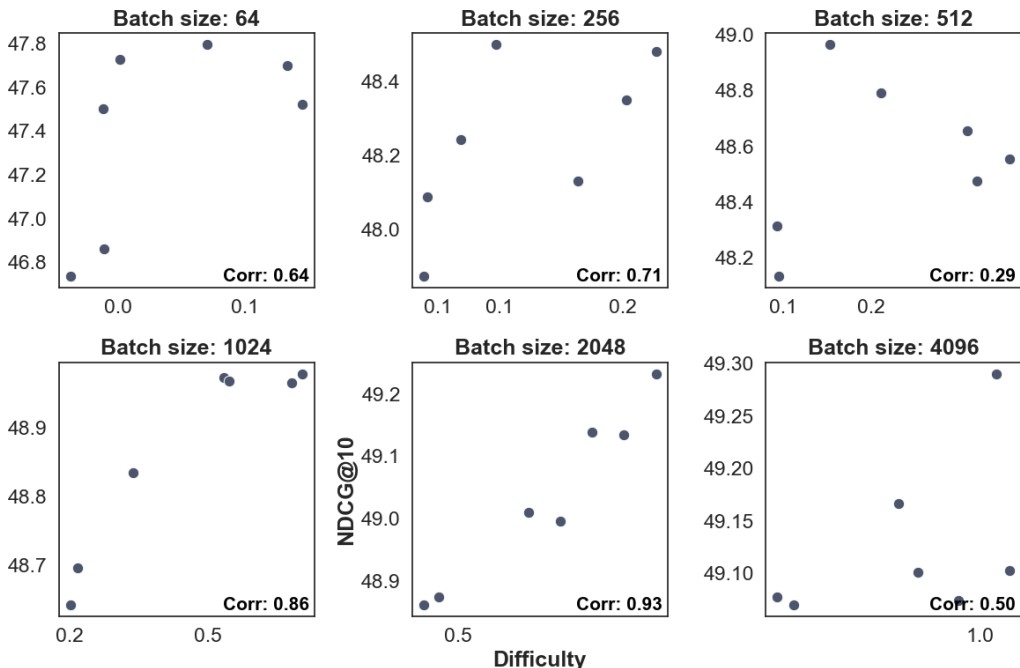

Figure 4: Correlation between batch difficulty and performance after supervised training.

**Supervised training: difficulty correlations.** In Section 10.5 we plot the correlation between batch difficulty and downstream performance across cluster sizes (and within batch sizes) in the supervised setting. In this case we also see the best performance through the most difficult clusters.

**Supervised training: full results.** We plot the full results of all supervised training experiments in Section 10.5. Our experiments in this setting (using the mined negatives from the Nomic supervised meta-datasets) generally show *decreasing* performance with additional hard negatives.

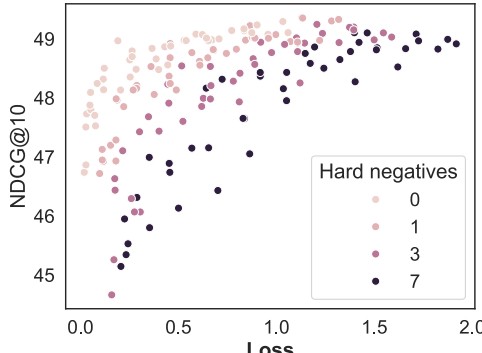

Figure 5: Performance of all supervised models, across numbers of hard negatives.

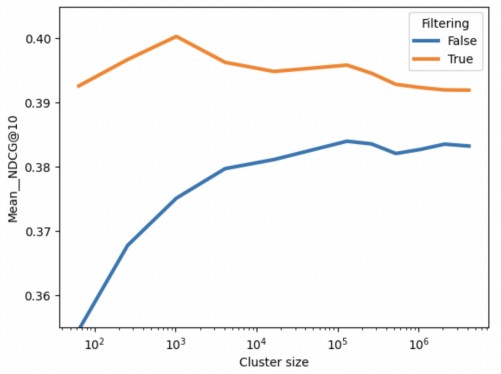

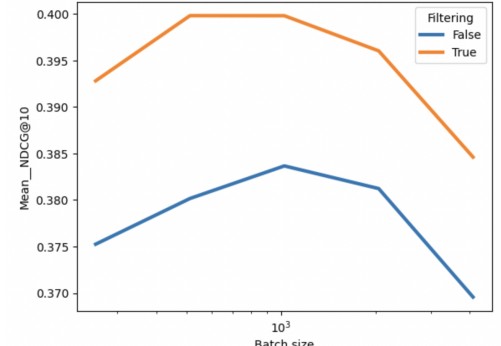

Figure 6: Model performance vs. cluster size with and without filtering. When false negative filtering is enabled, we see more improvements in performance from clustering at small cluster sizes.

Figure 7: Model performance vs. batch size with and without filtering. With and without filtering, the optimal batch size ranges between $10^2$ and $10^4$; performance starts to decrease as batch size grows too large.

**TSP Packing.**  We compare randomly packing clusters into batches vs. a greedy traveling salesman-style solution, similar to (Shi et al., 2024). In our scenario, we first cluster datapoints, then find the centroid embedding of each cluster. We begin packing by randomly selecting a cluster, and then choose the next cluster by finding the cluster with the closest centroid to the current one. Results are shown in Figure 8. Although these results appear slightly noisy, we see an improvement from TSP-style packing especially at smaller cluster sizes (where packing has an outsized impact). We therefore opt to use this packing procedure for our main model.

**Impact of context size**  We consider contextual embeddings might move in space as their conditioning varies. Section 10.5 displays a few qualitative examples. We generate embeddings for randomly sampled documents from the TREC-Covid dataset and visualize their embeddings with PCA, where unique document inputs with different contextual embeddings are visualized in the same color. By changing only the conditioning we reshape the embedding space and our model produces different embedding for the same text. Note that although the embeddings are clearly moving in response to changing the contextual inputs, they still remain closer to each other than to different documents.

We also consider how additional context is improving our model. Because the model includes an optional null token, we can supply any number of contextual inputs. We plot our model's performance across context sizes in Figure 10.5. We see that our model is able to utilize partial context window sizes, and even perform reasonably with no context (i.e. all null token inputs) but offers the best performance given a full context window size.

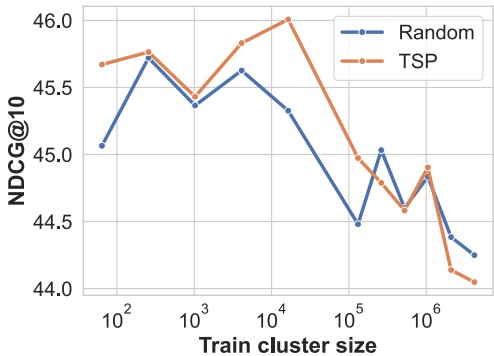

Figure 8: Pre-training with TSP vs. random batching across cluster sizes.

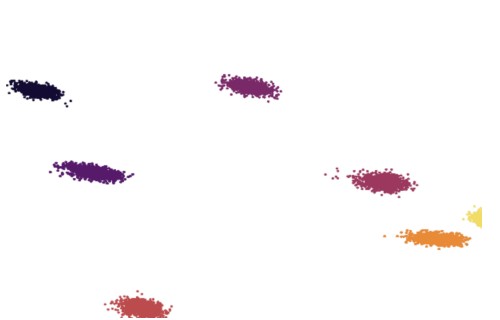

Figure 9: Each color indicates a single document input $d$. Different points represent different values $\phi(d; \mathcal{D})$ for different contexts.

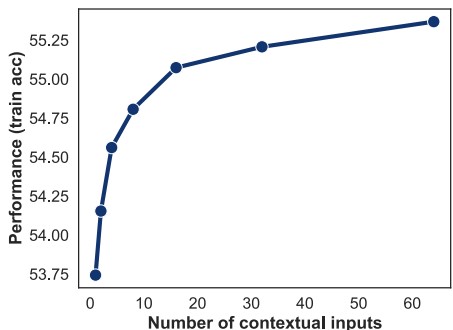

Figure 10: Performance of CDE model as the number of contextual examples increases.

### 10.6 CLUSTER TEXT EXAMPLES

icinnblkkcrdjejtltglnkcrtbbhekuvuvbjdhgtrbjffennbfrtcgckthuvt

We include random examples from a cluster gathered from our supervised dataset, shown in Table 6. This particular cluster appears to be a combination of documents about county populations in the Untied States (in Kentucky, Iowa, Pennsylvania, etc.) and documents about criminal trials (mentioning hearings, depositions, and courts).

### 10.7 TASK PREFIXES

Prefixes are hand-written for each dataset in both meta-training sets. We follow the same prefix selection procedure as Nussbaum et al. (2024), inspired by Reimers et al. (2023):

- `search_query`

- `search_document`

- `classification`

- `clustering`

| query | document |
|---|---|
| population of breckenridge mi | breckenridge, michigan. breckenridge is a village in gratiot county in the u. s. state of michigan. the population was 1, 328 at the 2010 census. the village is located in wheeler township. |
| can a deposition be used in a criminal case | depositions are commonly used in civil litigation (suits for money damages or equitable relief) [...] |
| what cases require strict scrutiny | the strict scrutiny standard is one of three employed by the courts in reviewing laws and government policies. the rational basis [...] |
| function of state supreme courts | it has also initiated several programs designed to improve the effectiveness of the court system. a primary function of the supreme court is to ensure [...] |
| what is the population in idaho | idaho ' s population grows to nearly 1. 7 million. idaho ' s population grew by 1. 2 percent between mid - 2014 and mid - 2015, the 12th strongest increase among the states and four - tenths of a percentage point ahead of the national growth rate. |
| what is the population of manson, ia | manson, iowa. manson is a city in calhoun county, iowa, united states. the population was 1, 690 at the 2010 census. |
| what happens after a sentencing hearing | find answers. sentencing. after a criminal defendant is convicted or pleads guilty, a judge will decide [...] |
| flathead county population | flathead county, montana. flathead county is a county located in the u. s. state of montana. as of the 2010 census, the population was 90, 928, making it [...] |
| whiting, ks population | the city of whiting had a population of 177 as of july 1, 2017. whiting ranks in the lower quartile for population density and diversity index when compared to the other cities, towns [...] |
| what is the population of lewiston id | lewiston, id population and races. as of 2010 - 2014, the total population of lewiston is 32, 178, which is 4. 12% more than it was in 2000. [...] |
| what happens if you don't show up for jury | what happens if you don't show up for jury duty in california? a : according to california courts, judicial branch of california, if a citizen fails to show up for jury duty, the juror can accrue fines up to $1,500. if service presents an undue hardship, a juror can request a postponement or to be excused. otherwise, citizens are not exempt from jury duty. |
| population of clearfield county pa | clearfield is a borough and the county seat of clearfield county, pennsylvania, united states. the population was 6, 215 at the 2010 census, and the borough is part of the dubois, pa micropolitan statistical area, as well as the larger state college - dubois, pa combined statistical area. |
| how long can it take for a trial | the preliminary hearing phase of the trial usually takes place 5 - 6 days after an arraignment. in the case of a misdemeanor [...] |
| population clinton ky | clinton county is a county located in the u. s. state of kentucky. as of the 2010 census, the population was 10, 272. its county seat is albany. the county was formed in 1835 and named for dewitt clinton, the seventh governor of new york. it is a prohibition or dry county. |
| population of iosco county michigan | with 25, 420 people, iosco county is the 55th most populated county in the state of michigan out of 83 counties. but watch out, iosco county, because gladwin county with 25, 411 people and manistee county with 24, 420 people are right behind you. |

Table 6: Sixteen samples from a cluster our algorithm finds in the supervised training data. The full cluster size is 256 points out of a dataset of $1.5M$.

Table 7: Distribution of pretraining datasets curated in Nussbaum et al. (2024).

| Dataset | Datapoints | % Dataset |
|---|---|---|
| Reddit[a] | 64,978,944 | 0.28 |
| PAQ Lewis et al. (2021) | 52,953,088 | 0.23 |
| Amazon Reviews Ni et al. (2019) | 38,682,624 | 0.16 |
| S2ORC Title Abstract Lo et al. (2020) | 35438592 | 0.15 |
| WikiAnswers Fader et al. (2014) | 9,912,320 | 0.04 |
| S2ORC Citation Titles Lo et al. (2020) | 7,585,792 | 0.03 |
| S2ORC Abstract Citation Lo et al. (2020) | 7,503,872 | 0.03 |
| S2ORC Abstract Body Lo et al. (2020) | 6,389,760 | 0.03 |
| Wikipedia Title Body Foundation (2024) | 6,078,464 | 0.03 |
| Gooaq Khashabi et al. (2021) | 1,245,184 | 0.01 |
| Codesearch Husain et al. (2019) | 835,584 | <.01 |
| AGNews ? | 409,600 | <.01 |
| CCNews Hamborg et al. (2017) | 344,064 | <.01 |
| NPR[b] | 344,064 | <.01 |
| CNN See et al. (2017) | 278,528 | <.01 |
| Yahoo Title-Answer[c] | 262,144 | <.01 |
| AmazonQA Gupta et al. (2019) | 212,992 | <.01 |
| Yahoo Title-Question[d] | 196,608 | <.01 |
| Sentence Compression Filippova & Altun (2013) | 163,840 | <.01 |
| YahooQA[e] | 131,072 | <.01 |
| ELI5 Fan et al. (2019) | 98,304 | <.01 |
| Altlex Hidey & McKeown (2016) | 98,304 | <.01 |
| Wikihow Koupaee & Wang (2018) | 81,920 | <.01 |
| SimpleWiki Coster & Kauchak (2011) | 81,920 | <.01 |
| StackExchange Duplicate Questions[f] | 65,536 | <.01 |
| StackExchange Title Body[g] | 65,536 | <.01 |
| StackExchange Body Body[h] | 65,536 | <.01 |
| Quora Duplicate Questions[i] | 32,768 | <.01 |
| SQuAD Rajpurkar et al. (2016) | 16,384 | <.01 |
| Total | 234,553,344 | 1 |

[a] https://huggingface.co/datasets/sentence-transformers/reddit-title-body

[b] https://files.pushshift.io/news/

[c] https://www.kaggle.com/soumikrakshit/yahoo-answers-dataset

[d] https://www.kaggle.com/soumikrakshit/yahoo-answers-dataset

[e] https://www.kaggle.com/soumikrakshit/yahoo-answers-dataset

[f] https://data.stackexchange.com/apple/query/fork/1456963

[g] https://data.stackexchange.com/apple/query/fork/1456963

[h] https://data.stackexchange.com/apple/query/fork/1456963

[i] https://quoradata.quora.com/First-Quora-Dataset-Release-Question-Pairs

Table 8: Distribution of BEIR evaluation datasets used, ordered by corpus size.

| Dataset | Queries | Documents |
|---|---|---|
| NFCorpus | 323 | 3,633 |
| SciFact | 300 | 5,183 |
| ArguAna | 1,406 | 8,674 |
| SciDocs | 1,000 | 25,657 |
| TREC-COVID | 50 | 171,332 |
| Quora | 5,000 | 522,931 |
| Natural Questions | 3,452 | 2,681,468 |
| MS MARCO | 6,980 | 8,841,823 |

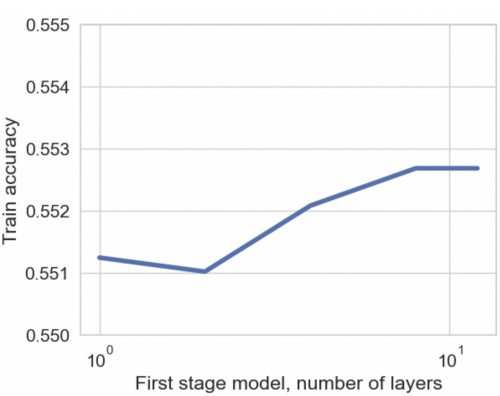

Figure 11: System performance (training accuracy) as we scale the size of the first-stage model encoder only.

## 10.8 UNSUPERVISED TRAINING DATASETS

We train on $234M$ weakly supervised query-document pairs collected for training text embedding models in Nussbaum et al. (2024). The full distribution of 29 datasets is shown in Table 7. Reddit alone makes up over 25% of the data distribution, with 19 of the datasets comprising under 1% of the total data.

## 10.9 BEIR EVALUATION DATASETS

Our initial experiments involve evaluating on nine datasets from the BEIR benchmark. Datasets are detailed in Table 8. To enable fast evaluation at this stage, we obtain the top 1024 relevant documents to each document with GTR (Ni et al., 2021) and rerank only these documents at evaluation time.

## 10.10 ADDITIONAL MODELING ABLATIONS

**First-stage model size.** One consideration is whether we can improve our system without affecting search inference time by scaling the number of parameters in the backbone model only. We study this affect by scaling the number of layers in the transformer backbone of the first-stage model from 1 to the full 12. Resulting performance is shown in Section 10.10.

Our results show that scaling the first-stage model has a small positive influence on model performance. However, since the total improvement from a 12x increase in first-stage model size is less than one percent, we conclude that the second-stage model size has a much larger impact on performance.

## 10.11 HOW MANY TOKENS PER DOCUMENT?

We consider the question of how many tokens per document is ideal while keeping the total number of document tokens fixed. Results per the nine evaluation datasets of BEIR are shown in Section 10.11.

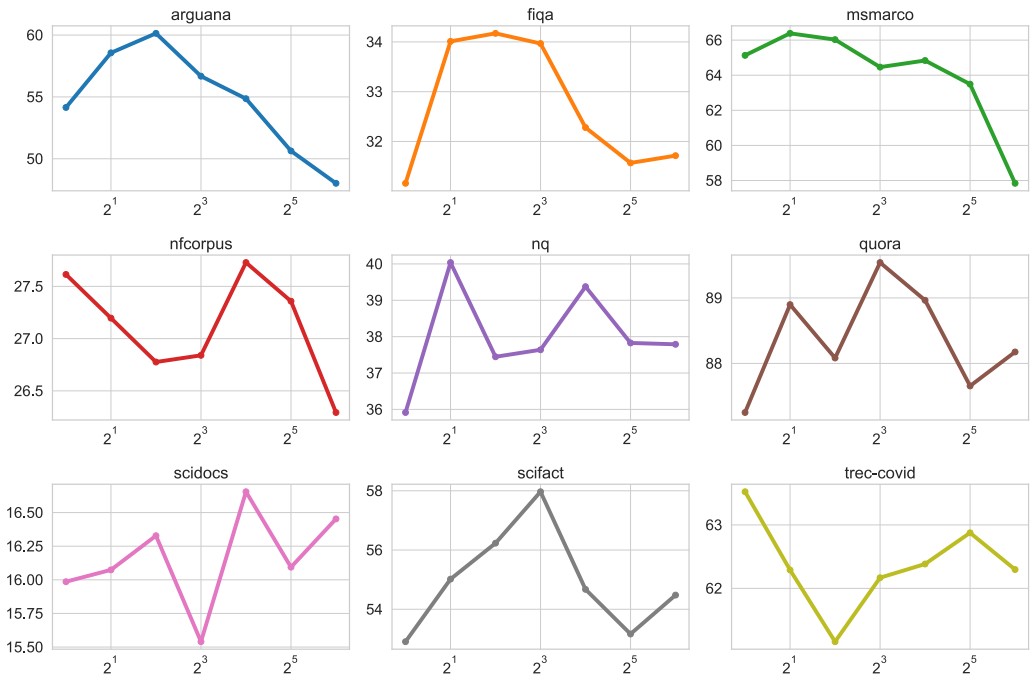

Figure 12: Performance per-dataset as we scale tokens-per-document, while keeping the total number of contextual tokens fixed. Different domains prefer a different number of tokens per document.

## 10.12   MTEB RETRIEVAL EVALUATION PERFORMANCE

| | Random | Contextual |
|---|---|---|
| AmazonCounterfactualClassification | 87.6 | 87.0 |
| AmazonPolarityClassification | 94.6 | 94.7 |
| AmazonReviewsClassification | 55.7 | 55.8 |
| ArguAna | 66.5 | 72.0 |
| ArxivClusteringP2P | 47.1 | 48.6 |
| ArxivClusteringS2S | 38.1 | 40.5 |
| AskUbuntuDupQuestions | 60.2 | 61.3 |
| BIOSSES | 86.2 | 86.7 |
| Banking77Classification | 87.5 | 88.6 |
| BiorxivClusteringP2P | 41.5 | 44.9 |
| BiorxivClusteringS2S | 35.8 | 37.9 |
| CQADupstack | 38.0 | 39.8 |
| ClimateFEVER | 19.2 | 25.7 |
| DBPedia | 39.4 | 40.1 |
| EmotionClassification | 74.9 | 74.9 |
| FEVER | 81.6 | 86.1 |
| FiQA2018 | 40.7 | 45.4 |
| HotpotQA | 63.8 | 66.7 |
| ImdbClassification | 93.4 | 93.6 |
| MSMARCO | 40.9 | 41.5 |
| MTOPDomainClassification | 95.4 | 96.0 |
| MTOPIntentClassification | 86.8 | 89.2 |
| MassiveIntentClassification | 74.8 | 76.6 |
| MassiveScenarioClassification | 77.8 | 78.9 |
| MedrxivClusteringP2P | 34.6 | 37.7 |
| MedrxivClusteringS2S | 33.0 | 35.5 |
| MindSmallReranking | 30.4 | 30.1 |
| NFCorpus | 34.0 | 34.7 |
| NQ | 57.1 | 60.0 |
| QuoraRetrieval | 89.0 | 89.8 |
| RedditClustering | 59.0 | 59.4 |
| RedditClusteringP2P | 64.1 | 64.5 |
| SCIDOCS | 19.2 | 21.8 |
| SICK-R | 77.0 | 77.4 |
| STS12 | 74.3 | 74.2 |
| STS13 | 85.6 | 86.1 |
| STS14 | 78.4 | 78.9 |
| STS15 | 87.4 | 87.5 |
| STS16 | 84.3 | 84.3 |
| STS17 | 89.2 | 89.7 |
| STS22 | 66.1 | 66.0 |
| STSBenchmark | 85.0 | 85.5 |
| SciDocsRR | 80.1 | 82.9 |
| SciFact | 71.4 | 71.8 |
| SprintDuplicateQuestions | 95.2 | 95.3 |
| StackExchangeClustering | 66.0 | 66.8 |
| StackExchangeClusteringP2P | 38.2 | 38.9 |
| StackOverflowDupQuestions | 50.6 | 52.7 |
| SummEval | 31.6 | 31.2 |
| TRECCOVID | 76.1 | 79.4 |
| Touche2020 | 29.4 | 24.3 |
| ToxicConversationsClassification | 74.4 | 72.8 |
| TweetSentimentExtractionClassification | 72.7 | 72.6 |
| TwentyNewsgroupsClustering | 55.3 | 56.9 |
| TwitterSemEval2015 | 71.6 | 72.9 |
| TwitterURLCorpus | 85.6 | 85.9 |
| Mean | 63.8 | 65.0 |

Table 9: Performance of our model with contextual documents (*"Contextual"*) and without (*"Random"*) on all the tasks of MTEB.

To evaluate on MTEB, we subsample contextual documents from the full corpus available in each dataset and modality. For retrieval, this corresponds to the corpus itself (importantly, not the queries); for other modalities, we choose the default "text" field in each casel. For classification tasks, we sample from the text side (not the classification labels themselves).

Table 9 shows our model performance on all datasets in the MTEB retrieval category. We see largest improvements over the baseline on the ArguAna and TREC-Covid datasets.

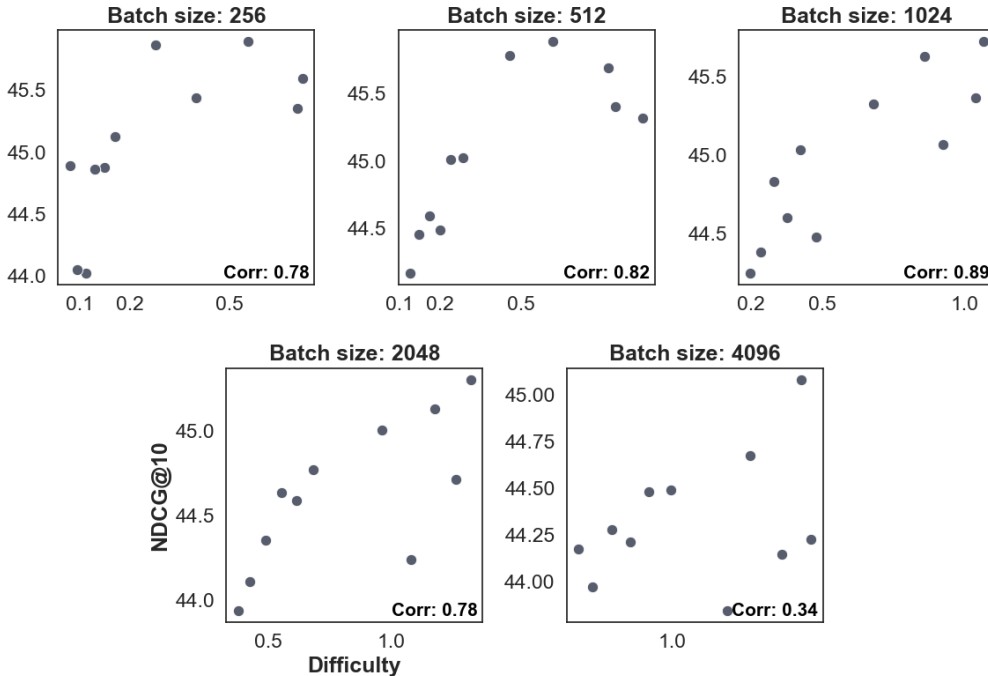

Figure 13: Performance vs. average batch difficulty (as measured by loss at the end of pre-training and supervised training) across batch sizes, after supervised contrastive training. Within a given batch size, we observe a clear increase in performance by making individual batches harder. Correlations are Pearson.

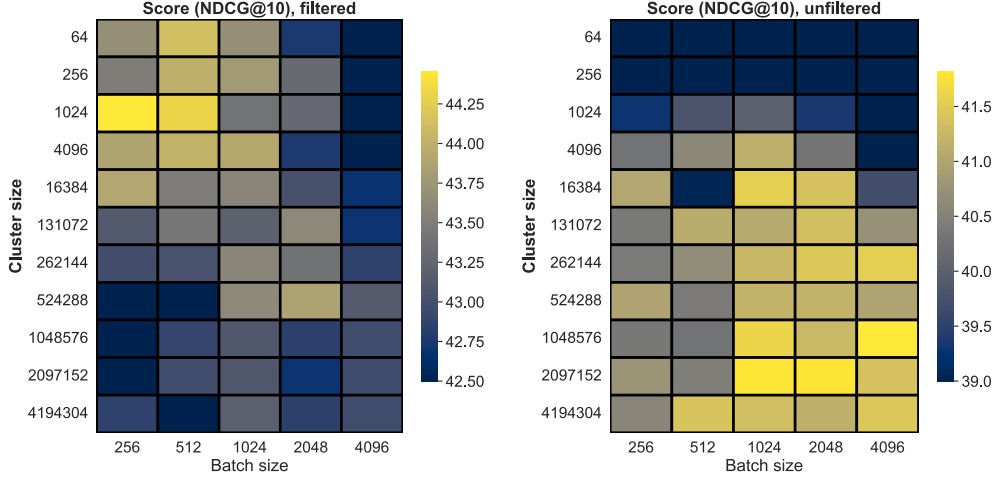

Figure 14: Biencoder performance with filtering (left) and without (right) across batch and cluster sizes during unsupervised contrastive pre-training. With filtering, small cluster sizes clearly improve performance, and larger batch sizes do not.

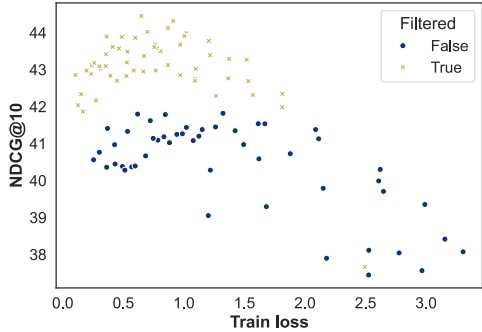

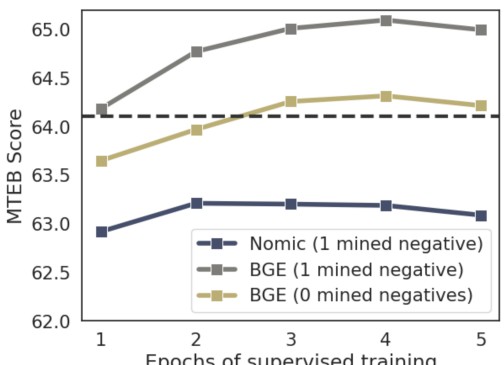

Figure 15: Impact of filtering during training across various batch and cluster sizes. Each dot is a biencoder pretrained with a different batch and cluster size.

Figure 16: Performance on MTEB across epochs of supervised training on the Nomic and BGE supervised meta-datasets.

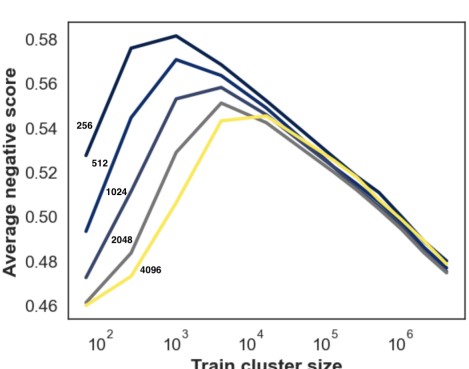

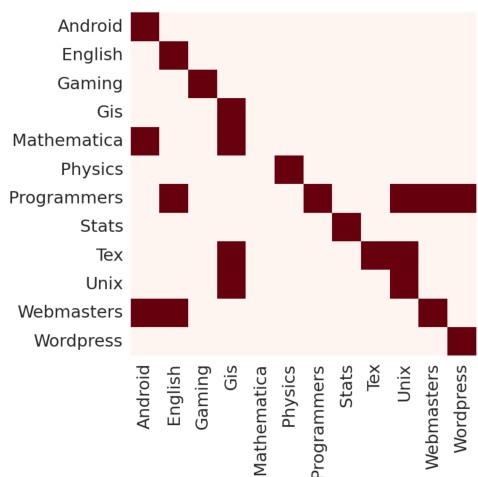

Figure 17: Average difficulty of in-batch negatives as measured by a surrogate model as cluster size and batch size change.

Figure 18: Impact of context by testing our model with different Stackexchange forum input types. Y-axis indicates the input domain, X-axis indicates the test domain. Dark squares come within one point NDCG@10.