# OpenReview forum: "Contextual Document Embeddings"
_ICLR.cc/2025/Conference — ICLR 2025 Poster_

### Official Review · Reviewer_Ja4L · 2024-10-29

**Soundness:** 3
**Presentation:** 3
**Contribution:** 4
**Rating:** 8
**Confidence:** 4

**Summary:**

The paper explores contextual document embeddings by factoring in the information from surrounding documents in context in order to effectively handle retrieval tasks in out-of-domain scenarios that can be extremely challenging for existing dense neural encoders.

The authors propose two complementary methods to improve the dense neural encoders in such settings: a) contrastive learning objective to explicitly incorporate neighboring documents into the contextual batch loss b) architectural modification to explicitly encode information from neighboring documents.

The results on MTEB benchmark demonstrate SOTA performance without the need of sophisticated strategies like hard negative mining, intra-GPU example sharing or large batch sizes which are often employed by prior works. The proposed method is agnostic and can be applied to any contrastive learning dataset or dual encoder models.

**Strengths:**

1. The proposed method incorporates corpus-level statistics through statistical retrieval method into dense neural biencoders to better contextualize the embeddings by conditioning on the neighboring documents. This effectively address the lack of contextual information not captured by existing neural biencoders and makes the contextualized document embeddings to be particularly beneficial in domain-specific retrieval tasks.

2. Most of the prior works on training dense neural biencoders often employ strategies like hard negatives or large batch sizes to achieve effective performance. It is interesting that the proposed method achieves competitive or superior performance without relying on such strategies.

3. The authors conduct extensive ablation studies to understand the impact of different components on the downstream performance, and these findings will be valuable for future research in this area.

4. The paper mostly list all the hyperparameters, datasets and models used in this work, ensuring the reproducibility of work.

**Weaknesses:**

1. The proposed method relies on the availability of relevant document neighbors for contextualization in the second stage of proposed architecture. The authors provide some evidence (Figure 7) on how the model performance varies in scenarios with limited context by simulating using random documents. However, it is not clear how the model trained with the proposed method would perform in absence of context (using null tokens in place of contextual tokens). Does the method makes the model overly reliant on context?

2. Following on the previous point, the contextual tokens might need to be augmented with the null tokens and this may potentially lead to unnecessary computational overhead.

3. The effectiveness of the proposed method heavily depends on accurately clustering documents into fine-grained pseudo-domains using GTR model, as these clusters are later used for contextual training. I am uncertain if using a simpler embedding model would result in obtaining difficult batch configurations, as it might fail to account for nuances. Do you have any ablations with alternative embedding models?

4. The authors mention that they set the dataset input tokens to zero during the self-attention phase in the section 9.4 in the appendix ("position-agnostic embedding" para in the manuscript). However, self-attention plays a critical role in contextualizing and routing necessary information to the subsequent modules. I am not sure how contextualization will be achieved if this is true. I would appreciate clarification from the authors.

**Questions:**

See weaknesses section for questions and concerns.

Few suggestions below:

1. Equation on page 4 in the manuscript uses symbol "m" in two different contexts which could make the notation a bit confusing. It might be better to use distinct symbols to avoid ambiguity.

2. The contextual architecture resembles to be somewhat similar to RETRO in terms of inspiration. It would be useful to add some discussion in the section 4.2.

3. Section 6 mentions that using single hard negative per query achieves better performance compared to without using any hard negatives. The authors should also report the results for both variants in Figure 5.

---

> ### Author Response · Authors · 2024-11-15
>
> Thank you for the thoughtful and thorough review. Here are our responses:
>
> > However, it is not clear how the model trained with the proposed method would perform in absence of context (using null tokens in place of contextual tokens). Does the method make the model overly reliant on context?
>
> We originally provided results with Random Documents as contextual inputs as evidence for our model’s performance without insight onto the test domain. We saw a decrease in average MTEB performance from 65.0 to 63.8, a difference of 1.2 points. We have re-tested with null tokens and achieved an MTEB score of 64.1, which indicates that the null token used during training is a more useful substitute for a contextual token than picking a random training document. Thank you for the suggestion.
>
> > Following on the previous point, the contextual tokens might need to be augmented with the null tokens and this may potentially lead to unnecessary computational overhead.
>
> We would like to clarify that the “null token” is a vector with parameters learned through gradient descent and included in the tokens passed to our model when context isn’t available. That means that there is a computational cost to our model even when context is not available. However, we do see our model with null tokens still outperforms a traditional biencoder; we think of this as a type of prompt tuning, where our model learns to use the extra computation provided by the null tokens to produce better downstream representations.
>
> > The effectiveness of the proposed method heavily depends on accurately clustering documents into fine-grained pseudo-domains using GTR model, as these clusters are later used for contextual training. I am uncertain if using a simpler embedding model would result in obtaining difficult batch configurations, as it might fail to account for nuances. Do you have any ablations with alternative embedding models?
>
> We took your suggestion and trained four of our small (32-token) biencoders with different clustering methods:
> gtr-base [Original]
> bert-base
> sbert (all-MiniLM-L12-v2)
> No clustering
>
> Here are the BEIR results:
> | Clustering  Model | BEIR NDCG@10 |
> |-------------------|--------------|
> | n/a               | 31.1         |
> | GTR               | 34.3         |
> | BERT-base         | 33.8         |
> | SBERT             | 33.1        |
>
> We saw all clustering settings outperformed no clustering, and GTR was the most effective. We were surprised to see SBERT slightly underperform BERT-base.
>
>
> > The authors mention that they set the dataset input tokens to zero during the self-attention phase in the section 9.4 in the appendix ("position-agnostic embedding" para in the manuscript). However, self-attention plays a critical role in contextualizing and routing necessary information to the subsequent modules. I am not sure how contextualization will be achieved if this is true. I would appreciate clarification from the authors.
>
> This is a great question! And was, truly, a writing error on our part. The dataset input tokens are set to zero *only during the rotary embedding part* of the transformer forward pass. If our model used position embeddings we could just disable those. But since it uses RoPE, we have to manually intervene and prevent the contextual tokens from undergoing RoPE.
>
> Notably, we *do* include contextual tokens in self-attention – otherwise, as you point out, our method wouldn’t work at all. We edited 9.4 to clarify these details.
>
> > Equation on page 4 in the manuscript uses symbol "m" in two different contexts
>
> This is a confusing detail, thank you for the feedback. We’ve renamed both m’s to different letters, M for “mean squared error” and c for “centroid”.
>
> > The contextual architecture resembles to be somewhat similar to RETRO…
>
> This is an interesting point – we had mentioned RETRO in the related work (the non-parameteric modeling section) but not directly in Section 4.2. We added another citation to RETRO in 4.2 along with a short discussion of its relevance.
>
> > Section 6 mentions that using single hard negative per query achieves better performance compared to without using any hard negatives. The authors should also report the results for both variants in Figure 5.
>
> We actually did already do this, although it was certainly not clear. The gray line uses one hard negative and the yellow line doesn’t use any. We will make this especially clear in Section 6.

---

> > ### Comment · Reviewer_Ja4L · 2024-11-19
> > **Rebuttal Acknowledgement**
> >
> > I would like to thank the authors for providing the clarifications and addressing my concerns and questions. I wish to maintain the same rating and look forward to revisions in the final version.

---

### Official Review · Reviewer_2RGZ · 2024-11-04

**Soundness:** 3
**Presentation:** 4
**Contribution:** 4
**Rating:** 8
**Confidence:** 4

**Summary:**

The paper proposes two improvements for document embedding models: (1) using clustering to group similar documents together to make contrastive batches more difficult, and (2) making document embedding contextual through an alternate architecture that takes in not only the document itself, but also neighboring documents. Both improvements lead to better performance on document retrieval benchmarks.

**Strengths:**

(1) Overall, the paper works on an important problem and provides two clean improvements that seem well-motivated and effective.

(2) The paper is well-written and clear: each section is well-organized, the methods are well-motivated, and each aspect of the method is explained clearly.

(3) The experiments are thorough and include useful ablations and analysis.

**Weaknesses:**

(1) The contextual architecture seems more expensive than its non-contextual counterpart, so the experiments would be clearer if they also reported wall clock training time. For example, from what I understand, all four methods in table 1 were trained for the same number of steps; would the results change if they were trained for the same amount of time instead?

(2) The paper states that hyperparameters were chosen based on the small-scale experiments, but later also states "For our final model (anon-model-v1), we select the best of the supervised models, which comes from finetuning on the BGE dataset." What is the set of models you are selecting from, and what metric are you using to perform selection?

(3) The paper mentions that the "largest improvements [were] in ArguAna and SciFact, two of the smaller and more out-of-domain datasets," but I wasn't able to find a breakdown of the BEIR results for individual datasets in Table 1 or the supplementary material.

**Questions:**

See above.

---

> ### Author Response · Authors · 2024-11-15
>
> Thank you for your helpful comments. We tried to resolve experimental questions below:
>
> > The contextual architecture seems more expensive than its non-contextual counterpart ... from what I understand, all four methods in table 1 were trained for the same number of steps; would the results change if they were trained for the same amount of time instead?
>
> Training our contextual model takes exactly twice as long as training a traditional biencoder, since we encode all documents with both $M_1$ and $M_2$ the same number of times. In practice we see the contextual model learns slowly and continues to increase performance beyond that of a biencoder. If the two models were trained for the same amount of time under our settings, we might expect similar performance; it takes the full multi-epoch training for the contextual model to see significant gains.
>
> > What is the set of models you are selecting from, and what metric are you using to perform selection?
>
> This is a great question and something not originally mentioned in the paper. We select the best batch size and cluster size from all of the small models (every colored box in Figure 3). The best parameters are batch size 256 and cluster size 1024, so we use these settings for the final model (anon-small-v1). We will add this information to our final manuscript.
>
> > I wasn't able to find a breakdown of the BEIR results for individual datasets in Table 1 or the supplementary material.
>
> We have conducted a more thorough analysis of this for our full model on the entire MTEB comparing performance with contextual and non-contextual documents. We observe that contextual inputs help most on ClimateFEVER (+6.5 NDCG), ArguAna (+5.5), and FiQA (+4.7) - see the general response for all data. (We will also add this new table to the appendix of the newest version of our paper. Thank you for the suggestion!)

---

> > ### Comment · Reviewer_2RGZ · 2024-11-21
> >
> > Thanks for the detailed answers.

---

### Official Review · Reviewer_zMkE · 2024-11-04

**Soundness:** 3
**Presentation:** 3
**Contribution:** 3
**Rating:** 6
**Confidence:** 3

**Summary:**

The paper proposed a batch sampling and architecture change to allow for contextual document embeddings. They achieve state-of-the-art results on the MTEB benchmark.

They embed a document in a two-stage approach. They first use clustering on embeddings generated by existing text embedding model to form a batch. Then they generate a single embedding by taking both the input document and the context documents in the same batch.

**Strengths:**

- A new batch sampling technique.
  - State-of-the-art on the MTEB benchmark.

**Weaknesses:**

Even though the paper is motivated by adapts the model to out-of-domain corpus, it's not evaluated on the domain-shift paradigm.

**Questions:**

- The paper motivates the work by introducing context into document embeddings. However, the context is collected by clustering and therefore it is more similar to increase the difficulty of the in-batch negatives.
- I also don't fully get how the batch of documents can capture the domain shift at inference time.

---

> ### Author Response · Authors · 2024-11-15
>
> Thank you for the helpful points, especially the feedback about out-of-domain evaluation; we have conducted a new experiment to explore these properties of our model further. Please see our General Response comment for new experimental data.
>
> > [The model] is not evaluated on the domain-shift paradigm.
>
> This is a very good point. To answer your question, we conducted an experiment where we compared the performance of our model with corpus-specific contextual document tokens vs. generic documents sampled from the training corpus. We saw striking increases in performance in settings that would imply domain shift: the top three datasets from MTEB with increased contextual performance were ClimateFEVER (climates about document), ArguArguana (“argument evidence”), and
>
> Thanks to your feedback, we were able to conduct more thorough analysis of where and how our model performs better out of domain (see General response for full data). In particular, contextual inputs help most on ClimateFEVER (+6.5 NDCG), ArguAna (+5.5), and FiQA (+4.7). These are some of the most out-of-domain datasets in MTEB – containing climate-related evidence, financial documents, and academic arguments. These are also some of the most out-of-domain datasets according to our IDF shift metric (see Appendix, 10.2, “Initial Experiments”).
>
> > the context is collected by clustering and therefore it is more similar to increase the difficulty of the in-batch negatives.
>
> This is true. Partitioning the training data into clusters of similar datapoints by definition increases the average difficulty of in-batch negatives. However, the process is much cheaper, since we don’t select negatives for every single query-document pair using a reranker, we instead run a biencoder once for each query & document and do dataset-level clustering. We will clarify this in the final version.
>
> > I also don't fully get how the batch of documents can capture the domain shift at inference time.
>
> We subsample 512 documents from the test corpus at inference time, which we assume to be enough to provide a high amount of context about the test dataset. An ablation study (Figure 16) showed diminishing returns when providing above even 16 contextual inputs. We leave it to future work to more thoroughly examine the effect of this number and build models that incorporate more or less of the test corpus at search time.

---

> > ### Comment · Reviewer_zMkE · 2024-11-25
> >
> > I would like to thank authors for their reply and clarification. I increased my score based on the reply and comments from other reviewers.

---

### Official Review · Reviewer_2sbA · 2024-11-05

**Soundness:** 3
**Presentation:** 3
**Contribution:** 2
**Rating:** 6
**Confidence:** 3

**Summary:**

The paper hypothesizes that dense embeddings lose the benefit of statistical embedding approaches in using corpus statistics, and proposes a two-step method to leverage the corpus information (e.g., documents in the same domain). The method first uses an extra embedding model to rearrange the training set by using their embeddings for clustering, such that similar documents can be in same batches for contrastive learning; in inference time, it further uses similar documents to get embeddings contextualized on these given documents.

**Strengths:**

1. The method is well-motivated, although it is unclear whether without the method, current state-of-the-art embedding models are not able to provide good nuanced representations.
2. The authors interpret the training of dense embedding methods and the method itself from a statistical perspective which is convincing.

**Weaknesses:**

The authors claim that no hard negative mining is required to achieve state-of-the-art. However, the first step of the method (grouping similar documents) is essentially hard negative mining and is shown to be a key contribution to the performance. At the end, it is mentioned that an extra hard negative per query is used to achieve the best performance.

**Questions:**

1. Can the authors provide more information about how the model is trained in the two-stage pass? Also, is the final model M1 or M2, or the both of them?
2. When evaluating on MTEB, how the in-context documents are selected for each task in the second-stage encoding?

---

> ### Author Response · Authors · 2024-11-15
>
> Thank you for the helpful review.  Here are our responses:
>
> >  It is unclear whether without the method, current state-of-the-art embedding models are not able to provide good nuanced representations.
>
> We show in the appendix (Initial Experiment, 10.2) how BM25 relatively outperforms dense methods when the corpus is dissimilar to the training corpus. Our new experiments (see General Response comment) indicate that this is no longer true for contextual embeddings. For example, although ArguAna is one of the more unusual datasets according to our IDF distance metric, incorporating contextual documents increases our ArguAna score by 5.5 NDCG@10.
>
> > Authors claim that no hard negative mining is required … an extra hard negative is [still] used to achieve the best performance.
>
> Everything you wrote is true, and a fair point. Although we were able to surpass the second-best model and achieve SOTA without any hard negatives (see Figure 5) – adding a single hard negative boosted performance even more. We will make this more clear in the final version, since the initial wording was indeed confusing, especially in the introduction.
>
> > Can the authors provide more information about how the model is trained in two stages?
>
> Our model is trained using a variant of the gradient caching technique, where M1 is run twice – once frozen, to get outputs, and another time unfrozen, to get gradients. We update the weights of both M1 and M2. Both models are updated on the same data batch using the same optimizer and learning ate. We edited 4.1 and 4.2 to make this information clear (especially under “two-stage gradient caching”).
>
> > Is the final model M1 or M2?
>
> The final model is M2. Once M1 is used to compute contextual embeddings, it can be discarded (and certainly doesn’t need to stay on GPU). We clarified this in 4.2.
>
> > When evaluating on MTEB, how the in-context documents are selected for each task in the second-stage encoding?
>
> Since the contextual size is rather large (512 documents) we assume that this is a representative sample of the full dataset and do simple random selection. We re-emphasized this detail to the experimental details section.

---

> ### Author Response · Authors · 2024-11-26
>
> Hello! Thanks again for the review of our paper. We'd like to request your response as the rebuttal period comes to a close. We added more information about our two-stage model training and the in-context document selection process. We also clarified our use of the term "hard negatives" as our contextual batching method is sort of a global hard-negative selection process.
>
> Please let us know if there any other concerns that we can answer or address before the rebuttal period ends – thank you.

---

> > ### Comment · Reviewer_2sbA · 2024-11-26
> >
> > thanks for the response. I am happy to increase the score. Just to note that the focus of the hard negative comment was not at all about the extra one hard negative that was used, but the fact that rearranging the corpus as done in the first step of the framework is essentially hard negative mining, as essentially diffucult negatives are arranged in the same batch for the contrastive loss, which is way more powerful than the extra one hard negative. Would love to hear the authors' thoughts on this if possible.

---

> > > ### Author Response · Authors · 2024-11-26
> > >
> > > Thanks for clarifying, and we completely agree that our contextual batching is similar in effect to hard negative mining.
> > >
> > > To clarify: we phrased things this way because our system is simpler. Traditional hard negative mining requires building an index and doing per-example retrieval, typically with a more powerful reranking model built-in to filter out hard negatives. Our contextual batching just requires embedding everything once (using any embedder) and running a fast clustering step.
> > >
> > > That said, contextual batching can certainly be thought of as a variant of hard negative mining, with two differences: it selects hard negatives at the batch level rather than the sample level, and requires less overhead, as discussed above.
> > >
> > > We will add a paragraph about this distinction to the discussion.

---

### Comment · Area_Chair_Gse8 · 2024-11-21
**Reminder: Please respond and update the score if necessary**

Dear Reviewers,

Kindly ensure that you respond proactively to the authors' replies so we can foster a productive discussion. If necessary, please update your score accordingly. We greatly appreciate the time and effort you’ve dedicated to the review process, and your contributions are key to making this process run smoothly.

Thank you,

AC

---

### Meta-Review · Area_Chair_Gse8 · 2024-12-21

**Metareview:**

The paper introduces a two-step method aimed at leveraging corpus information, particularly focusing on documents within the same domain. The first step involves using an auxiliary embedding model to rearrange the training data by clustering documents based on their embeddings. This allows for similar documents to be placed in the same batches during contrastive learning. In the inference phase, the method employs these related documents to generate embeddings that are contextualized by their similar counterparts.

This well-conceived approach significantly enhances dense neural biencoders by incorporating corpus-level statistics, thereby improving the contextualization of embeddings through a statistical retrieval strategy. It effectively overcomes the shortcomings of existing neural biencoders in capturing contextual nuances and is especially beneficial for domain-specific retrieval tasks. Impressively, the method achieves competitive or superior performance without resorting to traditional strategies such as using hard negatives or large batch sizes.

The paper is written and well-organized, clearly explaining its methodologies and findings. It is complemented by extensive experimental evaluations and ablation studies that provide valuable insights into the method's effectiveness. Additionally, the paper ensures reproducibility by meticulously detailing the hyperparameters, datasets, and models utilized in the research. Considering these strengths, I recommend the paper for acceptance.

**Additional Comments On Reviewer Discussion:**

All reviewers have expressed satisfaction with the authors' responses.

---

### Decision · Program_Chairs · 2025-01-22

Accept (Poster)